# Plumbagin Suppresses Breast Cancer Progression by Downregulating HIF-1α Expression via a PI3K/Akt/mTOR Independent Pathway under Hypoxic Condition

**DOI:** 10.3390/molecules27175716

**Published:** 2022-09-05

**Authors:** Supawan Jampasri, Somrudee Reabroi, Duangjai Tungmunnithum, Warisara Parichatikanond, Darawan Pinthong

**Affiliations:** 1Department of Pharmacology, Faculty of Science, Mahidol University, Bangkok 10400, Thailand; 2Department of Pharmaceutical Botany, Faculty of Pharmacy, Mahidol University, Bangkok 10400, Thailand; 3Department of Pharmacology, Faculty of Pharmacy, Mahidol University, Bangkok 10400, Thailand; 4Center of Biopharmaceutical Science for Healthy Ageing (BSHA), Faculty of Pharmacy, Mahidol University, Bangkok 10400, Thailand

**Keywords:** plumbagin, hypoxia-inducible factor-1α (HIF-1α), breast cancer, PI3K/Akt/mTOR pathway, MCF-7 cells

## Abstract

Hypoxia-inducible factor-1α (HIF-1α) is a major transcriptional regulator that plays a crucial role in the hypoxic response of rapidly growing tumors. Overexpression of HIF-1α has been associated with breast cancer metastasis and poor clinical prognosis. Plumbagin, the main phytochemical from *Plumbago indica*, exerts anticancer effects via multiple mechanisms. However, its precise mechanisms on breast cancer cells under hypoxic conditions has never been investigated. This study aims to examine the anticancer effect of plumbagin on MCF-7 cell viability, transcriptional activity, and protein expression of HIF-1α under normoxia and hypoxia-mimicking conditions, as well as reveal the underlying signaling pathways. The results demonstrate that plumbagin decreased MCF-7 cell viability under normoxic conditions, and a greater extent of reduction was observed upon exposure to hypoxic conditions induced by cobalt chloride (CoCl_2_). Mechanistically, MCF-7 cells upregulated the expression of HIF-1α protein, mRNA, and the VEGF target gene under CoCl_2_-induced hypoxia, which were abolished by plumbagin treatment. In addition, inhibition of HIF-1α and its downstream targets did not affect the signaling transduction of the PI3K/Akt/mTOR pathway under hypoxic state. This study provides mechanistic insight into the anticancer activity of plumbagin in breast cancer cells under hypoxic conditions by abolishing HIF-1α at transcription and post-translational modifications.

## 1. Introduction

Breast cancer is the most common type of malignancy in women. Clinically, approximately 90% of breast cancer patient deaths are the result of metastasis [1]. The development of breast cancer is linked intratumorally to hypoxia, which occurs when the oxygen supply and demand are imbalanced due to the rapid growth of a large solid tumor mass [2,3]. In response to hypoxic conditions, cancer cells adapt to the stress environment by upregulating a transcription factor called hypoxia-inducible factor-1α (HIF-1α), which plays a critical role in the growth, invasion, metastasis, and therapeutic resistance of breast cancer [3,4]. HIF-1α exerts an important function in cellular adaptation to hypoxia by activating multiple cancer signaling pathways, including phosphoinositide 3-kinase (PI3K)/Akt/mammalian target of rapamycin (PI3K/Akt/mTOR) and ERK/MAPK. The oxygen availability regulates HIF-1α stability, in which HIF-1α is hydroxylated under normoxic conditions by the oxygen sensor prolyl hydroxylases (PHD), leading to its polyubiquitylation by the von Hippel–Lindau (VHL) protein E3 ligase and degradation in the proteasome. Under hypoxia, PHD activity is inhibited, resulting in the stabilization of HIF-1α in the cytosol. Subsequently, the activated HIF-1α will translocate into the nucleus and form a heterodimer with HIF-1β to transactivate the transcription of target genes involved in tumor survival, metastasis, and angiogenesis, including vascular endothelial growth factor (VEGF) [5].

Overexpression of HIF-1α in primary breast tumor biopsies has been linked to an increased risk of metastasis, and is proposed as an adverse indicator of prognosis and diagnosis in patients [1]. As overexpressed HIF-1α has been correlated with treatment failures, it is considered to be a potential drug target for breast cancer therapy [4]. Over the past two decades, tremendous efforts have been made to develop HIF-1α inhibitors to prevent the spread of cancers. However, none of the HIF-1α inhibitors have been approved for the treatment of breast cancer patients in clinics due to insufficient effectiveness, toxicity, and lack of specificity [6]. Searching for novel agents targeting HIF-1α with greater efficacy and specificity is thus necessary to improve anticancer therapy in the future.

In recent years, plant-based medicines have drawn much attention for cancer treatments due to their promising efficacy and low toxicity compared with conventional chemotherapeutic drugs. *Plumbago indica* L. (Family: Plumbaginaceae), a medicinal plant native to Thailand, China, India, Indonesia, and other Asian countries, has been traditionally used for carminative, tonic elements, blood tonic, and anti-diarrhea treatment. Plumbagin (5-hydroxy-2-methyl-1, 4-naphthoquinone) is the major bioactive component found in the roots of *P. indica* with a wide range of biological properties, including anti-inflammatory, antioxidant, antifungal, antiprotozoal, and antibacterial activities [7,8]. The chemical structure is shown in Figure 1. Several studies have reported the pharmacological properties of plumbagin, including anticancer activities on various types of cancer cell lines. The anticancer mechanisms of plumbagin are mostly associated with its anti-proliferative and apoptotic-inducing effects [9,10,11,12,13,14,15]. However, the effect under the hypoxic conditions of plumbagin has not yet been investigated in breast cancer. As hypoxia is a key feature of breast cancer progression, the present study investigated the effects of plumbagin on HIF-1α expression under hypoxic conditions, and the potential molecular mechanisms in MCF-7 cancer cells.

## 2. Results

### 2.1. Cytotoxic Effects of Plumbagin on MCF-7 Cells

The cytotoxic effects of plumbagin on MCF-7 cells were examined by MTT assay. In normoxic conditions, cells were treated with plumbagin at various concentrations (1, 2, and 4 μM) for 24, 48, and 72 h. The results show that the viability of MCF-7 cells was reduced in a concentration-dependent manner (Figure 2A,C,E). The IC_50_ values were 2.63 ± 0.01, 2.86 ± 0.01, and 2.76 ± 0.01 μM at 24, 48, and 72 h, respectively (Figure 2B,D,F and Table 1). Plumbagin did not display a time-dependent cytotoxic effect on MCF-7 cells compared to the vehicle (0.1% DMSO). The IC_50_ values of the compound were comparable at all time courses of incubation.

### 2.2. Effects of CoCl_2_-Induced Conditions on HIF-1α Expression and MCF-7 Cell Viability

To investigate the effect of plumbagin under hypoxic conditions, cell viability was initially examined by treating with CoCl_2_, a well-known hypoxia mimetic agent, to determine its noncytotoxic concentration before treating with plumbagin. The MTT results show that CoCl_2_ reduced the viability of MCF-7 cells in a concentration- and time-dependent fashion (Figure 3A). The least cytotoxic concentration for CoCl_2_ is 150 at 24 h of incubation. As HIF-1α is a key regulator of hypoxia, its expression at the protein level was further determined following CoCl_2_ treatments. The results demonstrate that CoCl_2_ induced the expression of HIF-1α protein in a concentration- and-time-dependent manner (Figure 3B,C). In particular, 6-h exposure of 150 μM CoCl_2_ is the optimal condition to achieve the increase in HIF-1α protein expression compared with the control, without using cytotoxicity. Based on the results, a pretreatment of MCF-7 cells with 150 μM CoCl_2_ was chosen to mimic hypoxic conditions, followed by treatments with plumbagin to investigate the cytotoxicity under chemical-induced hypoxia. In Figure 3D, the results show that 24-h exposure to plumbagin reduced the viability of MCF-7 cells, with an IC_50_ value of 2.30 ± 0.02 μM, compared with the vehicle control in a concentration-dependent manner.

### 2.3. Plumbagin Inhibited HIF-1α mRNA and Protein Expression under Hypoxic Conditions

To examine the effect of plumbagin on HIF-1α under CoCl_2_-induced hypoxic conditions, the changes in expression of the HIF-1α protein were determined by Western blotting. The findings reveal that CoCl_2_ potently induced the expression of HIF-1α proteins compared with the normoxic group (Figure 4A). Interestingly, plumbagin, at the concentration of 4 μM, significantly downregulated HIF-1α protein expression compared with the untreated hypoxic group. The relative quantities of HIF-1α protein are shown in Figure 4B. We next asked whether a decrease in HIF-1α protein by plumbagin is modulated at the transcriptional level, and mRNA expression of HIF-1α was determined by qRT-PCR. It was found that CoCl_2_ significantly increased the mRNA levels of HIF-1α compared to under normoxic conditions. While plumbagin at concentrations of 2 and 4 μM significantly downregulated the HIF-1α mRNA expression compared to CoCl_2_-induced untreated group (Figure 4C). These findings indicate that plumbagin exerts an inhibitory effect on HIF-1α by blocking its transcription and translation under hypoxia-induced conditions.

### 2.4. The mRNA and Protein Expression of HIF-1α Target Genes, Including VEGF-A and VEGFR-2, Were Downregulated by Plumbagin

We further assessed the suppressive effects of plumbagin on HIF-1α. The transcriptional activity and protein expression of its downstream targets were determined by q-PCR and Western blotting under CoCl_2_-induced hypoxic conditions in MCF-7 cells. The mRNA expression of the key HIF-1α target genes including *VEGF-A* and *VEGFR-2* was examined. The results demonstrate that plumbagin at the concentration of 4 μM significantly suppressed the mRNA expression levels of *VEGF-A* (Figure 5A). In comparison, *VEGFR-2* mRNA levels were reduced by plumbagin at concentrations of 2 and 4 μM (Figure 5B). However, it is interesting that VEGF-A protein expression was unaffected by plumbagin compared to both untreated normoxic and hypoxic groups (Figure 5C,D). Together, these results indicate that plumbagin blocks HIF-1α signaling by partly mediating the transcription of its target genes. 

### 2.5. Suppression of HIF-1α by Plumbagin Was Not Regulated by the PI3K/Akt/mTOR Pathway

The PI3K/Akt/mTOR signaling pathway is associated with the regulation of HIF-1α protein synthesis. To further examine the mechanism underlying HIF-1α inhibition by plumbagin, the PI3K, Akt, mTOR, and its phosphorylated forms were detected by Western blotting. BKM120 (PI3K inhibitor), MK2206 (Akt inhibitor), and rapamycin (mTOR inhibitor) were used as positive controls. Under CoCl_2_-induced hypoxia, the results show that rapamycin inhibited HIF-1α protein expression, similar to plumbagin compared with the mimic hypoxic group without treatment (Figure 6A,B). However, plumbagin did not affect PI3K, Akt, or mTOR in both phosphorylated forms and total protein levels (Figure 6C–H). The findings suggest PI3K/Akt-independent signaling pathways controlling the effect of plumbagin under CoCl_2_-induced hypoxia.

## 3. Discussion

This study reveals that plumbagin exerts anticancer activity in breast cancer cells by inhibiting HIF-1α activity under hypoxic conditions. Plumbagin reduced MCF-7 cell viability in a concentration-dependent manner under normoxic and hypoxic conditions. HIF-1α overexpressed under hypoxia is important for the adaptation of cancer cells to the low oxygen concentration of solid tumors within hypoxic regions. 

CoCl_2_, a chemical inducer of HIF-1α expression, mimicked hypoxia via the inhibition of hydroxylation by PHDs leading to HIF-1α stabilization, accumulation, and translocation into the nucleus. The advantages of the induction with CoCl_2_ are the ease of obtaining results and the inexpensive and time-efficient method used to induce hypoxia in cell culture, while the disadvantage of CoCl_2_ at high concentrations is that it can cause cytotoxicity [16]. A hypoxia incubator chamber can be used to create a hypoxic environment with 1% oxygen gas. The advantage is that 1% oxygen does not change cell behavior, while the disadvantages are expensiveness and difficulty to control reoxygenation during the experiment [17]. We demonstrated that HIF-1α expression levels were increased after exposure to CoCl_2_ at various concentrations and times. Incubation with CoCl_2_ at the concentration of 150 μM for 6 h is the optimal time and concentration, increasing both gene and protein expressions of HIF-1α. CoCl_2_ is used in several hypoxia induction studies in which it can induce HIF-1α expression at both mRNA and protein levels in many types of cancers, including MCF-7 cells [18,19,20,21,22]. Our results are in concordance with others, indicating that CoCl_2_ is an effective hypoxic inducer for in vitro experiments.

HIF-1α overexpression in solid tumors is associated with tumor progression via a variety of processes, including proliferation, angiogenesis, invasion, and metastasis [23,24,25]. In this study, we identified, for the first time, that plumbagin inhibited HIF-1α expression under hypoxic conditions, indicating that plumbagin has a translational inhibitory effect on HIF-1α. Furthermore, HIF-1α mRNA levels were measured to determine whether plumbagin affects HIF-1α at the translational or transcriptional level. We found that plumbagin significantly reduced HIF-1α expression at both the mRNA and protein levels, whereas several studies have reported effective inhibition of HIF-1α at the protein level. HIF-1α inhibition occurred as a result of a decrease in HIF-1α protein synthesis, which resulted in protein accumulation, but did not disrupt mRNA levels [26,27]. As a result, the precise underlying mechanisms of plumbagin on HIF-1α mRNA inhibition in MCF-7 cells under hypoxic conditions must be investigated further.

VEGF, a pro-angiogenic factor, is one of the downstream targets of HIF-1α critical for HIF-1α-mediated angiogenesis. As plumbagin downregulated HIF-1α protein expression, this event may subsequently lead to a decrease in the downstream targets of HIF-1α. This study provides evidence that plumbagin significantly downregulated mRNA expression of *VEGF-A* and *VEGF-R2*, key HIF-1α targets, indicating that plumbagin inhibited HIF-1α-mediated transcription in MCF-7 breast cancer cells under hypoxic conditions. It is well known that the VEGF-VEGFR-2 axis is a key signaling cascade of angiogenesis via the promotion of endothelial cell mitosis and chemotactic responses. Due to the overexpression of VEGF regulating the VEGFR-2 pathway via the phosphorylation of various tyrosine residues and the activation of multiple signaling cascades that are implicated in tumor angiogenesis [28,29]. Therefore, our results suggest that plumbagin might act as an anti-angiogenic agent by inhibiting the VEGF pathway. However, the protein expression of VEGF was unaffected by plumbagin treatment. This suggests that plumbagin may not be effective in modulating VEGF proteins under hypoxic conditions. The observed finding can be explained by the incomplete inhibition of VEGF proteins, which may result from the (1) post-transcriptional regulation of VEGF, including VEGF internal ribosome entry site (IRES)-mediated translation [30,31], microRNA (miRNA)-mediated regulation [32], or the regulation of cytokines, hormones, and growth factors [33,34]; or (2) post-translational modifications, such as phosphorylation, acetylation, or glycosylation, leading to VEGF protein stability [35].

The PI3K/Akt/mTOR signaling pathway is involved in regulating HIF-1α translation [36]. Previous research reported that plumbagin induced cell cycle arrest and autophagy via the inhibition of the PI3K/Akt/mTOR signaling pathway in MCF-7 cells [37]. Inhibition of HIF-1α expression also occurred via the suppression of the PI3K/Akt/mTOR signaling pathway in various cell types under hypoxia or CoCl_2_-induced hypoxia [38,39,40]. In the present study, we reveal that plumbagin did not affect the expression of total PI3K, Akt, or mTOR proteins, or their phosphorylated forms, under hypoxic conditions. The results suggest that plumbagin is unlikely to block HIF-1α signaling and its target genes by interrupting the PI3K/Akt/mTOR signaling pathway in MCF-7 breast cancer cells. However, the inhibition effect of plumbagin on HIF-1α may act through the inhibition of the ERK/MAPK-dependent signaling pathway (Figure 7). Further studies on other molecular mechanisms and targets of HIF-1α remain to be conducted in order to fully explain the inhibitory effect of plumbagin. In addition, MCF-7 cell lines have been shown to express high levels of HIF-1α protein [41]. Several studies found that plumbagin potently exhibited an anticancer effect against estrogen receptor (ER)-positive cell lines. Furthermore, plumbagin had a high specificity for ERs that were highly expressed in MCF-7 cells, but a low specificity for MDA-MB-231 cells [42]. Therefore, in this investigation, we aim to investigate the impact of plumbagin on the ER-positive MCF-7 cell line associated with HIF-1α. The impact of plumbagin on ER-negative MDA-MB-231 cells, however, is an intriguing subject that merits more research.

## 4. Materials and Methods

### 4.1. Chemicals and Reagents

Plumbagin, cobalt chloride (CoCl_2_), dimethyl sulfoxide (DMSO), 3-(4,5-dimethylthiazole-2-yl)-2,5-diphenyltetrazolium bromide (MTT), and bovine serum albumin (BSA) were purchased from Sigma-Aldrich (St. Louis, MO, USA). TRIzol was purchased from Invitrogen (Carlsbad, CA, USA). A protease inhibitor, BCA protein assay kit, polyvinylidene difluoride (PVDF) membrane, precision plus protein dual color standard, and nonfat dry milk were purchased from Bio-Rad (Hercules, CA, USA). A phosphatase inhibitor cocktail was purchased from Roche Diagnostic (Mannheim, Germany).

### 4.2. Cell Culture and Experimental Treatments

A human breast adenocarcinoma cell line (MCF-7) was obtained from American Type Culture Collection (ATCC; HTB-22, Manassas, VA, USA). Cells were maintained in complete growth media and Minimum Essential Medium (MEM) with Earle’s salt containing 10% fetal bovine serum (FBS), 100 μg/mL penicillin, and 100 μg/mL streptomycin (Capricorn Scientific, Ebsdorfergrund, Germany) at 37 °C in a humidified incubator with 5% CO_2_. Stock solutions of plumbagin and CoCl_2_ were dissolved in DMSO and stored at −20 °C. The final concentration of DMSO in the medium did not exceed 0.1%. Before experiments, MCF-7 cells were plated at the desired density in MEM supplemented with 10% FBS, and incubated at 37 °C in a humidified incubator with 5% CO_2_ for 24 h. Under the normoxic condition, cells were treated with plumbagin at various concentrations (1, 2, and 4 μM) for 24, 48, and 72 h in a quadruplicate manner. While under hypoxic conditions, cells were pretreated with 150 μM CoCl_2_ for 6 h, followed by treatment with plumbagin at various concentrations (1, 2, and 4 μM) in MEM supplemented with 1% FBS for a further 24 h in quadruplicates.

### 4.3. Cell Viability

The effect of plumbagin on cell viability under normoxic and hypoxic conditions was determined using an MTT assay. MCF-7 cells were plated into 96-well plates at a density of 1 × 10^4^ cells/well for 24 h, and then treated with various conditions according to the experimental design. After incubation, MTT working solution was added and incubated for 4 h. The medium was discarded and replaced with DMSO to solubilize the formazan product. The optical density was measured at a wavelength of 562 nm using a Varioskan Flash Multimode Reader (Thermo Fisher Scientific, Marietta, OH, USA). The cell viability of treatment groups was expressed as a percentage compared with the DMSO control.

### 4.4. RNA Extraction and Quantitative Real-Time PCR (qRT-PCR)

MCF-7 cells were plated at a density of 1.5 × 10^6^ cells/dish in 60 mm tissue culture dishes for 24 h. After incubation, cells were treated in various conditions according to the experimental design. Total RNA was extracted with TRIzol reagent. The purity of the extracted RNA was determined using NanoDrop. cDNAs were synthesized using the iScript™ cDNA Synthesis Kit (Bio-Rad, Hercules, CA, USA) according to the manufacturer’s protocol. Real-time PCR of HIF-1α, VEGF and VEGFR-2 were performed with iTaq™ Universal SYBR^®^ Green Supermix (Bio-Rad, Hercules, CA, USA) and analyzed with ABI PRISM7500 Sequence Detection System and analytical software (Applied Biosystems, Carlsbad, CA, USA). Samples were analyzed in a triplicate manner and the expression ratio was normalized with GAPDH. The relative expression of each gene was calculated by the ^∆∆Ct^ method. The threshold cycle (Ct) number of each gene was normalized to the Ct of GAPDH. Fold changes (arbitrary units) were determined as 2^−∆∆Ct^. A panel of PCR primers was designed using NCBI/Primer-Blast and the sequences were listed in Appendix A.

### 4.5. Western Blotting

Treated cells were harvested and lysed with ice-cold RIPA buffer containing protease inhibitors cocktail at 4 °C for 30 min. The cell suspension was centrifuged at 12,000 rpm at 4 °C for 20 min, and the supernatant was collected for protein analysis using the BCA protein assay kit. Twenty micrograms of protein were separated by 10% SDS-PAGE and transferred onto 0.2 μM polyvinylidene fluoride membranes (EMD Millipore, MA, USA). The membranes were immunoblotted with the indicated antibodies, including anti-HIF-1α (H1alpha67) (Novus Biologicals, CO, USA), anti-VEGF (Santa Cruz Biotechnology, CA, USA), anti-phospho-Akt (S473), anti-phospho-Akt (T308), anti-Akt, anti-phospho-mTOR, anti-mTOR, and anti-β-Actin antibodies (Cell Signaling Technology, Danvers, MA, USA). Blots were incubated in 5% nonfat dry milk in TBS-T (Tris, NaCl, Tween 20 at pH 7.4) for 2 h at room temperature to block nonspecific protein binding, and then probed overnight at 4 °C with primary antibody. Membranes were then incubated with corresponding horseradish peroxidase (HRP)-conjugated secondary antibodies for 1 h at room temperature. The detection of the bands was developed using an enhanced chemiluminescence reagent (ECL) (Millipore Corporation, Waltham, MA, USA) and imaged with the ChemiDoc™ Touch Imaging system (Bio-Rad, Hercules, CA, USA). The densitometry analysis was performed using provided software (Image Lab).

### 4.6. Statistical Analysis

All data are presented as the mean ± SEM from three independent experiments. Statistical differences between groups were compared using ANOVA followed by Tukey’s post hoc test. All statistical analyses were performed using GraphPad Prism (version 6; GraphPad software). *p*-values < 0.05 and <0.01 were considered statistically significant.

## 5. Conclusions

In conclusion, our results indicate that plumbagin exerts anticancer activity on MCF-7 breast cancer cells by blocking HIF-1α expressions at both mRNA and protein levels. Moreover, it suppresses HIF-1α-mediated *VEGF-A* and *VEGF-R2* mRNA expression through the PI3K/Akt/mTOR-independent signaling pathway. Our findings provide new and insightful information regarding the anticancer mechanisms of plumbagin in breast cancer. This compound is a promising natural inhibitor that could potentially be used to target HIF-1α-overexpressing cancer cells or combine with other endocrine therapies to reduce drug resistance or recurrence of hormone-dependent breast cancer. 

## Figures and Tables

**Figure 1 molecules-27-05716-f001:**
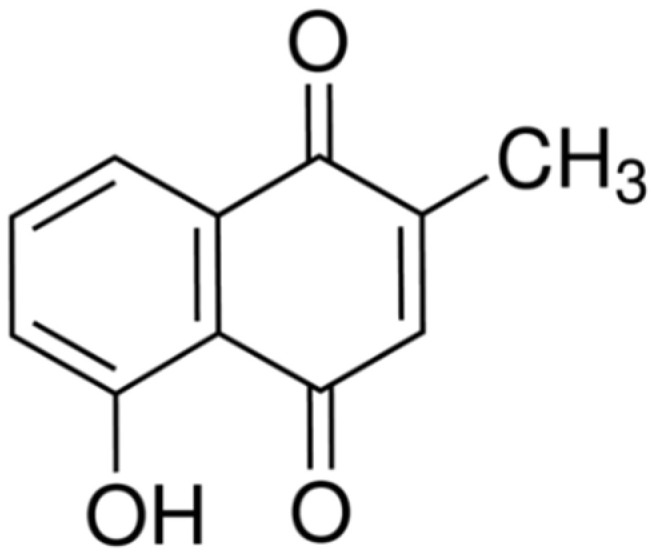
Chemical structure of plumbagin.

**Figure 2 molecules-27-05716-f002:**
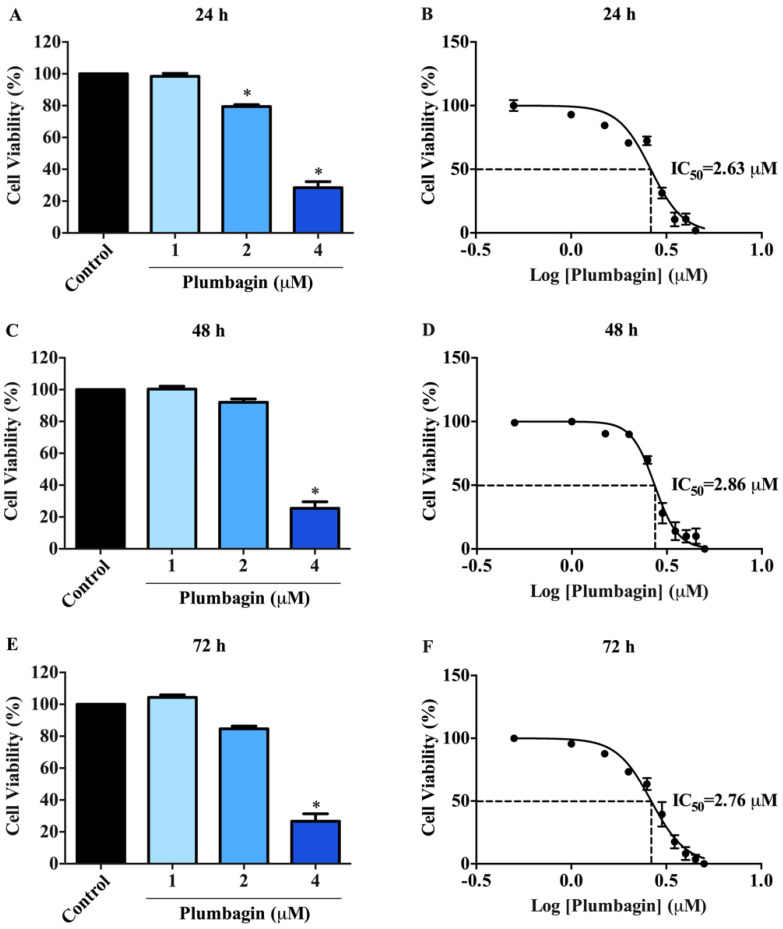
Cytotoxic effects of plumbagin on MCF-7 cells. MCF-7 cells were treated with 1, 2, and 4 μM of plumbagin for (**A**,**B**) 24 h, (**C**,**D**) 48 h, and (**E**,**F**) 72 h. The MCF-7 cell viability was measured by MTT assay. Data are means ± SEM compared with the control from three independent experiments (*n* = 3). * *p* < 0.05 vs. control.

**Figure 3 molecules-27-05716-f003:**
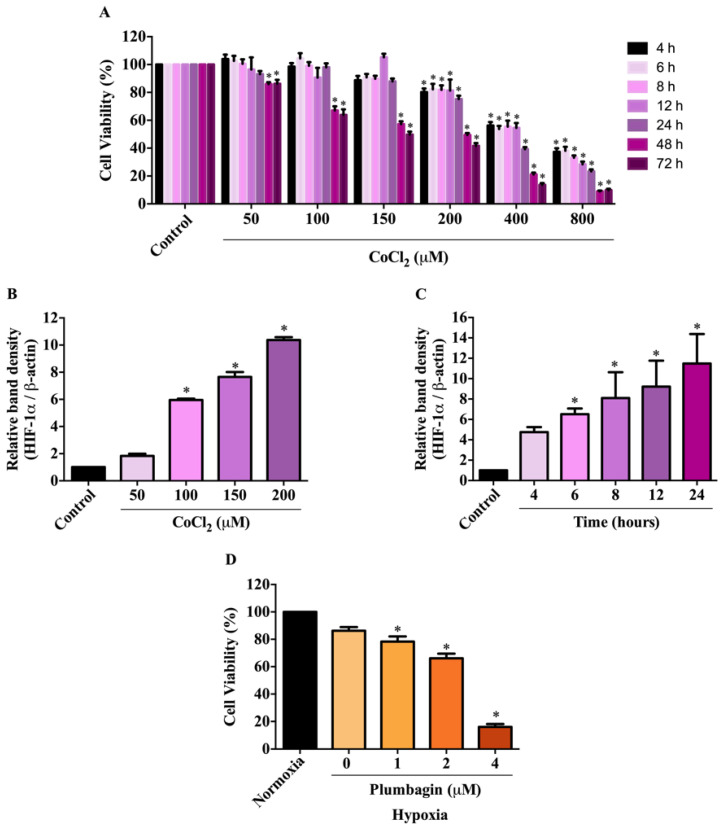
Effects of CoCl_2_-induced conditions on HIF-1α expression and MCF-7 cell viability. (**A**) MCF-7 cells were treated with 0–800 μM of CoCl_2_ for 4, 6, 8, 12, 24, 48, and 72 h and subjected to MTT assay. (**B**,**C**) HIF-1α protein expression of MCF-7 cells was analyzed by Western blotting after treatments with 50, 100, 150, and 200 μM of CoCl_2_ for 6 h or 150 μM of CoCl_2_ for 4, 6, 8, 12, and 24 h. The relative quantities of HIF-1α protein were quantified and normalized with β-actin. (**D**) MCF-7 cells were pretreated with 150 μM CoCl_2_ for 6 h and then treated with plumbagin for another 24 h. The MCF-7 cell viability was measured by MTT assay. Data are means ± SEM compared with the control from three independent experiments (*n* = 3). * *p* < 0.05 vs. control.

**Figure 4 molecules-27-05716-f004:**
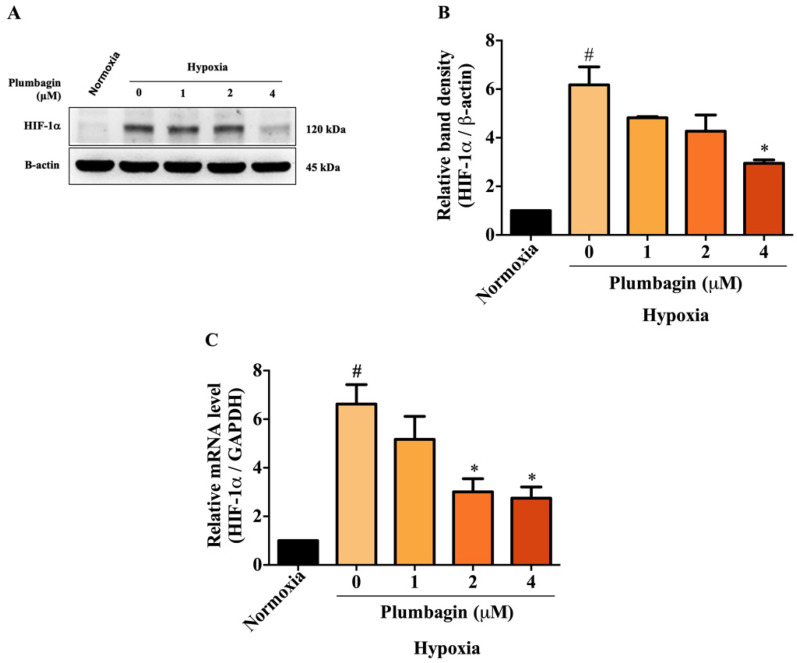
Plumbagin suppressed HIF-1α at transcriptional and translational levels under CoCl_2_-induced hypoxic conditions in MCF-7 cells. (**A**) HIF-1α protein expression was analyzed by Western blotting. MCF-7 cells were treated with 0–4 μM of plumbagin for 24 h under hypoxic conditions induced by CoCl_2_. (**B**) The relative quantities of HIF-1α protein were quantified and normalized with β-actin. (**C**) Plumbagin downregulated HIF-1α gene expression. Data are means ± SEM from three independent experiments (*n* = 3). # *p* < 0.05 vs. normoxic, * *p* < 0.05 vs. mimic hypoxic group without treatment.

**Figure 5 molecules-27-05716-f005:**
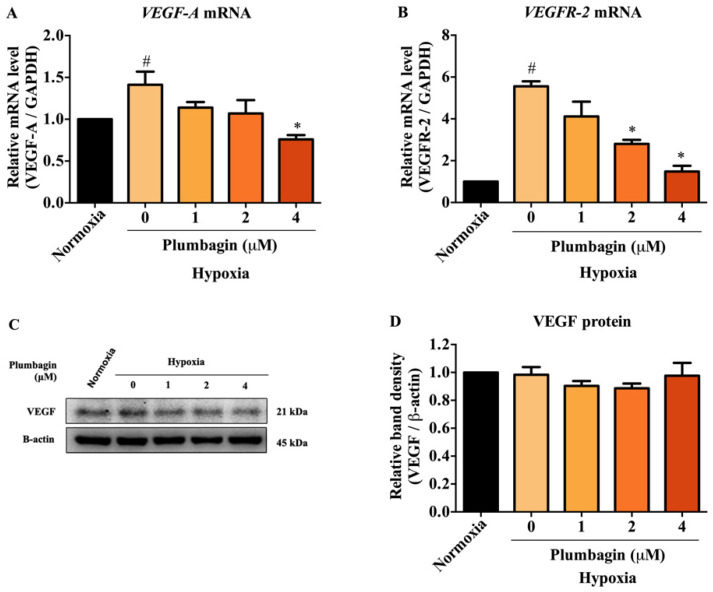
Plumbagin abolished mRNA and protein expression of HIF-1α target genes under hypoxia. (**A**,**B**) The mRNA expression of *VEGF-A* and *VEGFR-2* was determined by qRT-PCR. MCF-7 cells were treated with 1, 2, and 4 μM of plumbagin for 24 h under hypoxic conditions. The relative mRNA expression was quantified and normalized with GAPDH. Data are means ± SEM from three independent experiments (*n* = 3). # *p* < 0.05 vs. normoxic, * *p* < 0.05 vs. mimic hypoxic group without treatment. (**C**,**D**) The VEGF protein expression was investigated by Western blotting. MCF-7 cells were treated with 0, 1, 2, and 4 μM of plumbagin for 24 h under hypoxic conditions. The relative quantities of VEGF protein were quantified and normalized with β-actin. Data are means ± SEM from three independent experiments (*n* = 3).

**Figure 6 molecules-27-05716-f006:**
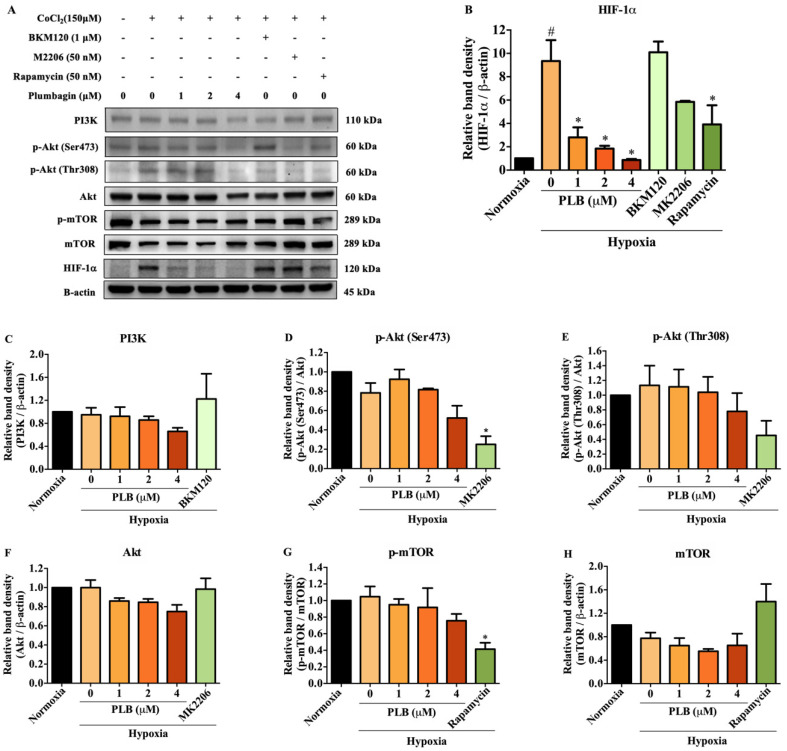
Plumbagin inhibited HIF-1α via PI3K/Akt/mTOR-independent signaling pathways under hypoxic conditions. (**A**) The protein expression of total PI3K, p-mTOR, and total mTOR were not altered by plumbagin. (**B**) Exposure to the compound downregulated HIF-1α protein expression. MCF-7 cells were treated with plumbagin (0–4 μM), BKM120 (1 μM), MK2206 (50 nM), or rapamycin (50 nM) under hypoxic conditions for 24 h and subjected to Western blotting. (**C**–**H**) The relative quantities of proteins were quantified and normalized with β-actin or total proteins. The relative quantities of p-Akt (Ser473), p-Akt (Thr308), and p-mTOR proteins were quantified and normalized with PI3K, Akt, or mTOR. Data are means ± SEM from three independent experiments (*n* = 3). # *p* < 0.05 vs. normoxic, * *p* < 0.05 vs. mimic hypoxic group without treatment.

**Figure 7 molecules-27-05716-f007:**
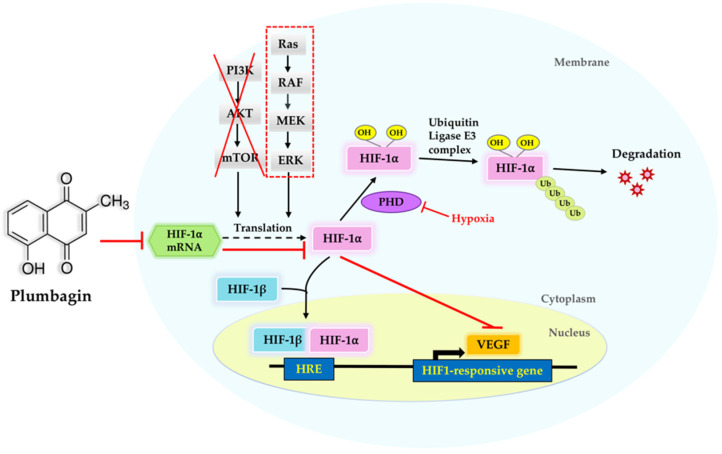
A proposed mechanism of action of plumbagin on MCF-7 cells.

**Table 1 molecules-27-05716-t001:** The IC_50_ values of plumbagin for MCF-7 cells with different exposure times.

Time (h)	IC_50_ ^a^ (µM)
24	2.63 ± 0.01
48	2.86 ± 0.01
72	2.76 ± 0.01

^a^ IC_50_ = Concentrations corresponding to 50% cell viability inhibition.

## Data Availability

Not applicable.

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
