# Peer review of "Plumbagin Suppresses Breast Cancer Progression by Downregulating HIF-1α Expression via a PI3K/Akt/mTOR Independent Pathway under Hypoxic Condition"

_molecules, 2022, doi:10.3390/molecules27175716_

Round 1

Reviewer 1 Report

1)      Authors should include more biological information on plumbagin in the introduction section. 

2)      Why the reported study is limited to only one breast cancer cell line (MCF-7)? Other breast cancer cell lines (MDA-MB-23) also need to be examined.

3)      Mechanism of action diagram (Figure 6) should move to the results and discussion and report the discussion accordingly.  

Author Response

Comments:

  1. Authors should include more biological information on plumbagin in the introduction section. 

Response: We would like to thank the reviewer for pointing out this issue. We added the information about the biological activities of plumbagin in the revised version of the manuscript (Page 2, line 69).

  1. Why the reported study is limited to only one breast cancer cell line (MCF-7)? Other breast cancer cell lines (MDA-MB-23) also need to be examined.

Response: In this study, MCF-7 breast cancer cell lines were chosen as the model for investigating the anticancer effect of plumbagin in hypoxic conditions because MCF-7 cell lines have been shown to express high levels of hypoxia-inducible factor-1α (HIF-1α) protein [1]. Several studies found that plumbagin potently exhibited anticancer effect against estrogen receptor (ER)-positive cell lines. Furthermore, plumbagin had a high specificity on ERs that were highly expressed in MCF-7 cells but a low specificity in MDA-MB-231 cells [2]. Therefore, in this investigation, we aim to investigate the impact of plumbagin on the ER-positive MCF-7 cell line associated with HIF-1α. The impact of plumbagin on ER-negative MDA-MB-231 cells, however, is an intriguing subject that merits more research.

References

  1. Shi, Y.; Chang, M.; Wang, F.; Ouyang, X.; Jia, Y.; Du, H. Role and mechanism of hypoxia-inducible factor-1 in cell growth and apoptosis of breast cancer cell line MDA-MB-231. Oncol. Lett. 2010, 1, 657-662.
  2. Kilcar, A.Y.; Tekin, V.; Muftuler, F.Z.B.; Medine, E.I. 99mTc labeled plumbagin: estrogen receptor dependent examination against breast cancer cells and comparison with PLGA encapsulated form. J. Radioanal. Nucl. Chem. 2016, 308, 13–22.

We added this information in the revised version of the manuscript (Page 8, line 259 and Page 13, line 462).

  1. Mechanism of action diagram (Figure 6) should move to the results and discussion and report the discussion accordingly.  

Response: We edited the number of figure 6 and moved the mechanism of action diagram (Figure 7) to the discussion part according to the relevant discussion in the revised version of the manuscript (Page 9, line 268).

Reviewer 2 Report

In this article Supawan el.al., and colleagues have studied the effect of Plumbagin, the main phyto-chemicals from Plumbago indica extracts, on hypoxia inducible factor -1α (HIF-1α) in only MCF-7 breast cancer cell line which is estrogen receptor (ER) and progesterone receptor (PR) positive.

Authors have shown reduction in HIF-1α level at RNA as well as protein level induced by CoCl2 after treatment with Plumbagin and correlated the survival data with the regulation of HIF-1α. The elucidation of mechanism is still unclear as has been said in the manuscript. The manuscript is well written and experiments are meticulously planned.

It is imperative to see if the effect of Plumbagin is similar on triple negative breast cancer cell line to conclude the claimed effect of Plumbagin on breast cancer cells.

Studies have shown that some plant derived natural compounds show quite different effect on different doses. So the in-vivo experiments will help to understand the final effect.

Author Response

 We would like to thank the reviewer for insightful suggestions.

The following are the reasons why MCF-7 breast cancer cell lines were chosen as the model for investigating the anticancer effect of plumbagin in hypoxic conditions in this study:

MCF-7 cell lines have clearly been reported to demonstrate high levels of hypoxia-inducible factor-1α (HIF-1α) protein expression [1]. Several studies revealed that plumbagin potently exhibited anticancer effect against estrogen receptor (ER)-positive cell lines. Moreover, plumbagin had high specificity on ERs that were highly expressed in MCF-7 cell, while their specificity was low in MDA-MB-231 cells. [2]. Therefore, we would like to examine the effect of plumbagin on ER-positive MCF-7 cell line correlated with HIF-1α in this study.

However, the effect of plumbagin on ER-negative MDA-MB-231 cells is an interesting topic that should be investigated further.

 References

  1. Shi, Y.; Chang, M.; Wang, F.; Ouyang, X.; Jia, Y.; Du, H. Role and mechanism of hypoxia-inducible factor-1 in cell growth and apoptosis of breast cancer cell line MDA-MB-231. Oncol. Lett. 2010, 1, 657-662.
  2. Kilcar, A.Y.; Tekin, V.; Muftuler, F.Z.B.; Medine, E.I. 99mTc labeled plumbagin: estrogen receptor dependent examination against breast cancer cells and comparison with PLGA encapsulated form. J. Radioanal. Nucl. Chem. 2016, 308, 13–22.

We added this information in the revised version of the manuscript (Page 8, line 259 and Page 13, line 462).

Studies have shown that some plant derived natural compounds show quite different effect on different doses. So the in-vivo experiments will help to understand the final effect.

In addition, we have agreed with the valuable comment from the reviewer that in vivo experiments are necessary to study further and clarify the anticancer efficacies and safety of plumbagin against breast cancer. Therefore, the findings from this study will be served as valuable information for further study of plumbagin in experimental animal models and in clinical studies.